# Application of decision analytic modelling to cardiovascular disease prevention in Sub-Saharan Africa: a systematic review

James Odhiambo Oguta [1] ✉, Penny Breeze[1], Elvis Wambiya[1], Peter Kibe[2], Catherine Akoth[1], Peter Otieno[2] & Peter J. Dodd [1]

## Abstract

**Background** This systematic review sought to examine the application of decision analytic models (DAMs) to evaluate cardiovascular disease (CVD) prevention interventions in sub-Saharan Africa (SSA), a region that has experienced an increasing CVD burden in the last two decades.
**Methods** We searched seven databases and identified model-based economic evaluations of interventions targeting CVD prevention among adult populations in SSA. All articles were screened by two reviewers, data was extracted, and narrative synthesis was performed. Quality assessment was performed using the Philips checklist.
**Results** The review included 27 articles from eight SSA countries. The majority of the studies evaluated interventions for primary CVD prevention, with primordial prevention interventions being the least evaluated. Markov models were the most commonly used modelling method. Seven studies incorporated equity dimensions in the modelling, which were assessed mainly through subgroup analysis. The mean quality score of the papers was 68.9% and most studies reported data challenges while only three studies conducted model validation.
**Conclusions** The review finds few studies modelling the impact of interventions targeting primordial prevention and those evaluating equitable strategies for improving access to CVD prevention. There is a need for increased transparency in model building, validation and documentation.

## Plain language summary

Cardiovascular Disease (heart disease) is an increasing problem in countries in sub-Saharan Africa. There are strategies in place to prevent disease and this review examined how mathematical tools for decision making are used to calculate how well prevention strategies are working. We performed a review of the literature on this topic and included 27 studies from eight SSA countries. We found common decision models used in many of the studies and very few studies with equity considerations (fairness to all). Challenges with data quality and limited real-world testing to show how well these tools work in practice were also found. These findings highlight the need for better mathematical tools and a greater focus on preventive strategies that are fair to all to help reduce heart disease in this region and improve public health.

Cardiovascular diseases (CVDs) are the leading causes of non-communicable disease (NCD) morbidity and mortality globally[1,2]. Recent estimates indicate that CVDs (ischaemic heart disease [IHD], intracerebral haemorrhage and stroke) were the highest contributors of age-standardised disability-adjusted life years (DALYs) in 2022[2,3]. The NCD burden is higher in low- and middle-income countries (LMICs), which account for more than three-quarters of all NCD related deaths and more than four-fifths of the premature deaths (occurring before the age of 70) attributed to NCDs[1].

In sub-Saharan Africa (SSA), the NCD burden has increased over the last three decades from about 18.6% (of all DALYs) in 1990 to 29.8% in 2019[4]. As globally, CVDs are the major causes of NCD deaths in SSA and were responsible for 13% and 37% of all-cause and NCD-related mortality in 2019, respectively[5]. The rising burden of CVD and their risk factors in the SSA region can be attributed to the demographic and epidemiological

transitions, rapid urbanisation and lifestyle changes that have occurred in the past decades[6,7].

In order to reverse the trend of CVDs in SSA, there is a need for the adoption and scale-up of effective and high-impact prevention interventions. The three main approaches to CVD prevention include[8,9]: (1) Primordial prevention, which targets individuals without CVD risk and aims at maintaining a low CVD risk status; (2) Primary prevention, which focuses on individuals who already have increased CVD risk with the aim of avoiding the onset of CVD; (3) Secondary prevention that targets individuals with CVD and aims at preventing complications including recurrent CVD events. In a setting like SSA where health infrastructure is weak and health systems are traditionally built to provide interventions for communicable diseases, it is particularly important to identify interventions that are not only effective but cost-effective and equitable at scale. Moreover, it is

[1]Sheffield Centre for Health and Related Research, Division of Population Health, School of Medicine and Population Health, Sheffield, UK. [2]African Population and Health Research Center (APHRC), Nairobi, Kenya. ✉e-mail: mcogutajamo@gmail.com

important to examine the equity impact of such interventions to inform viable options for attaining universal health coverage (UHC) in SSA.

Decision analytic modelling (DAM) is a valuable tool that can help to evaluate the health, economic and equity impact of different interventions for CVD prevention to inform priority setting. DAM involves the synthesis of evidence from multiple sources and the application of relevant mathematical techniques and computer software to predict the long-term impact of implementing a particular intervention[10]. The use of DAMs allows for the extrapolation of intervention costs and impacts beyond the study periods. Different cohort and individual patient level DAM approaches are available for modelling the impact of public health interventions for NCDs, with the model choice dependent on the nature of the decision problem[11,12].

Three previous reviews related to this topic focused on identifying cost-effective interventions for CVD prevention interventions in LMICs[13–15]. With primary focus on synthesising cost effectiveness evidence, these reviews included studies of different methodologies, including economic evaluations that did not use DAMs. Similarly, another review specific to the SSA setting appraised the sources of data used in economic evaluation studies of different NCD interventions but also included non-DAMs[16]. Moreover, none of the studies examined the methods used in modelling equity dimensions in existing DAMs for CVD prevention.

Our review adds to this literature by focussing on the use of DAMs in modelling CVD prevention interventions in the SSA setting. This review appraises the characteristics and quality of existing DAMs, the types of prevention interventions modelled, how CVD progression was modelled, and approaches to incorporating equity impacts of interventions. The review also appraises the quality of existing DAMs using the Phillips et al. checklist[17] and identifies existing gaps for future modelling studies. The specific objectives of the review included: 1) to identify the CVD prevention interventions and policies for which DAMs have been applied in SSA and existing gaps; 2) to examine the structure and characteristics of DAMs for CVD prevention interventions and policies in SSA; 3) to examine how equity is incorporated in model-based economic evaluations of CVD prevention in SSA; and 4) To assess the quality and identify the gaps in existing model-based economic evaluations for CVD prevention in SSA.

## Methods
We used the Preferred Reporting Items for Systematic Reviews and Meta-Analyses (PRISMA) 2020 guidelines to conduct and report the review[18]. The systematic review protocol was registered on PROSPERO (CRD42023457106).

### Study eligibility criteria
The review sought to identify model-based economic evaluations of interventions and policies targeting cardiovascular disease prevention in SSA.

Decision analytic models were defined as studies applying mathematical modelling techniques to predict the impact of interventions or policy options either in terms of their cost or health outcomes. We excluded economic evaluations performed alongside clinical trials or observational studies that did not extrapolate their results beyond the study period. Model-based evaluations of interventions targeting primordial, primary, and secondary CVD prevention among adult populations in SSA countries were included.

To be eligible, the studies must have modelled adult CVD with established prevention strategies (coronary heart diseases, stroke, heart failure or their variants) as outcomes. Articles evaluating interventions targeting rheumatic heart disease (RHD) were excluded from the review because RHD is caused by *Streptococcus pyogenes* bacteria and tends to affect the younger age groups[19,20]. Only published articles in peer-reviewed journals, in the English language, were included in the review. As such, conference proceedings, dissertations, opinion pieces, descriptive studies and letters to the editor were excluded. We also excluded grey literature. Table 1 summarises the study inclusion and exclusion criteria.

### Literature search
An iterative process was used to develop the strategy involving review of existing systematic reviews of economic evaluation studies and identification of relevant synonyms, discussions with other members of the review team and consultation of an information specialist from the University of Sheffield library. The strategy was developed by combining the four parts of the review question using appropriate Boolean operators as follows:

(Decision analytic models OR synonyms) AND (cardiovascular disease OR synonyms) AND (prevention OR synonyms) AND (SSA OR SSA countries OR synonyms).

The initial search strategy was piloted in the MEDLINE database and reviewed by the team before being adapted to suit the other databases. The final search was performed in seven databases that include MEDLINE via Ovid, EMBASE, APA PsycInfo, Scopus, Web of Science, EconLit and CINAHL from inception until September 12, 2023. Hand searching of reference lists of existing reviews[13,15,21] was also done to identify additional references for inclusion in the review. Detailed search strategies for each of the databases are presented in Supplementary Methods.

### Study selection process
Search results were exported into the Endnote reference manager where duplicates were identified and removed. After deduplication, the references were converted into an Endnote XML file and imported into Covidence software, where additional duplicates were automatically removed prior to the screening. All titles and abstracts and full texts were screened by two

## Table 1 | Systematic Review Inclusion and Exclusion Criteria

| | Included | Excluded |
|---|---|---|
| Population | Adult population aged at least 18 years | Children |
| Intervention | Public health interventions targeting primordial, primary, and secondary prevention | Studies with no intervention explicitly stated; treatments and specialised procedures delivered within clinical settings. |
| Comparator | Varied depending on the type of intervention being evaluated. | Studies without comparators |
| Health Outcome | Cardiovascular diseases including coronary heart diseases (angina and myocardial infarction), stroke, cerebrovascular accidents, heart failure and other non atherosclerotic CVDs | Rheumatic heart disease and Congenital heart diseases |
| Setting | Sub-Saharan Africa | Global studies not reporting results specific to the sub-Saharan African context. |
| Outcomes reported | Health impact, equity outcomes, incremental cost effectiveness ratios | Costing studies, cost of illness studies, burden of disease studies |
| Types of evaluations | Decision analytic models e.g., decision trees, Markov models, microsimulations, systems dynamic models, agent-based models | Economic evaluations performed alongside clinical trials or observational studies with a short time horizon |
| Publication type | Peer-reviewed publications in journals | Grey literature |
| Language | English | Other languages |

reviewers (JO and any of EW, PK, and CA). Conflicts were resolved by a third reviewer, not among the two initial reviewers.

### Data extraction

An Excel-based data extraction tool was used to capture data on the most important elements of the studies. The data extracted included study characteristics, type of intervention, model type, CVD outcomes, risk equations used, data sources, uncertainty analyses, and equity analysis among others.

### Quality assessment

We used the Philips checklist to assess the quality of the included studies[17]. Each study was appraised based on the extent to which it met each element of the checklist. We assigned a score of 1 (Y) for each criterion that was fully met, 0.5 score (U) where the criterion was partially met. A score of zero (X) was assigned where the authors did not report or include required information against the dimension of the checklist. An element of the checklist was tagged as "not applicable(N/A)" where it was not relevant to the study being evaluated. The quality assessment was performed by JO and reviewed by EW, PK, CA, PB and PD.

### Data synthesis

A narrative synthesis was conducted to assess the DAMs for CVD prevention in SSA based on the identified criteria. The studies were first categorised based on their characteristics, settings and types of interventions and policies modelled. We then compared the studies based on how they approached the modelling of CVD progression, their equity considerations,

assumptions, and limitations. All statistical analyses were performed using R software (version 4.4.1). Results were presented in a narrative format. The extracted data were presented using tables and graphs.

### Reporting summary

Further information on research design is available in the Nature Portfolio Reporting Summary linked to this article.

## Results

Out of an initial 2033 results retrieved from the database search, the final review included 27 papers[22–48]. Figure 1 presents the PRISMA flow diagram.

### Characteristics of the included studies

Figures 2 and 3 present the characteristics of the included studies, with specific details presented in Supplementary Data 1.

Figure 2A presents the distribution of the studies by country. South Africa had the highest number (seven) studies[24,30,32,33,36,37,48] followed by Tanzania with four studies[38–40,43] while Nigeria had three[28,36,44]. Cameroon[22,23], Ethiopia[27,47], Ghana[29,42] and Kenya[34,46] had two studies each, while Uganda had one study[45]. In five studies, several LMICs were grouped together, and the impact of interventions or policies evaluated at regional or multicountry level[25,26,31,35,41]. All the studies were published after 2005, with the majority (20/27) being published after 2015 (Fig. 2B).

### Types of interventions evaluated

Regarding the level of CVD prevention, 13 studies[22,23,27–30,32–34,37–40,42,44–46,48] evaluated interventions targeting primary prevention, five studies[22,23,37,40,48]

**Fig. 1 | PRISMA Flow Diagram Depicting the Study Selection Process.** The PRISMA (Preferred Reporting Items for Systematic Reviews and Meta-Analyses) flow diagram outlines the study selection process. The numbers show the studies selected or excluded at each step of the study selection.

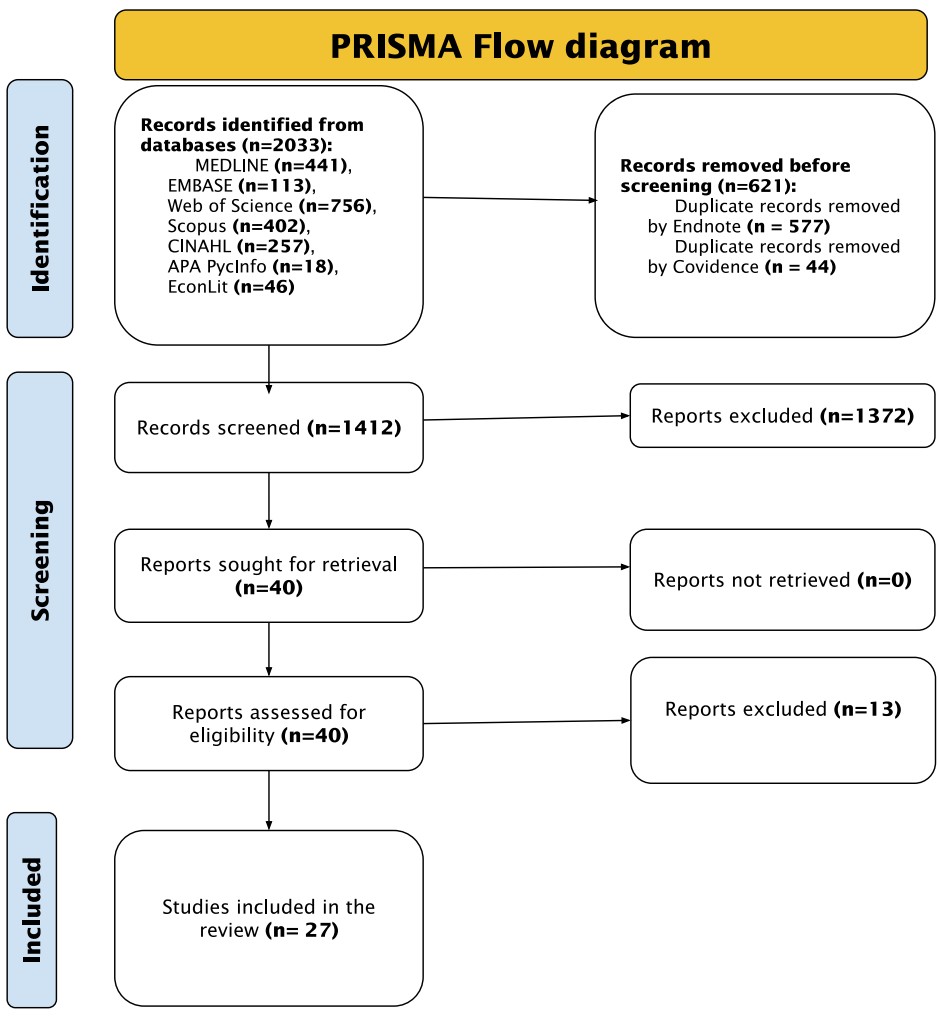

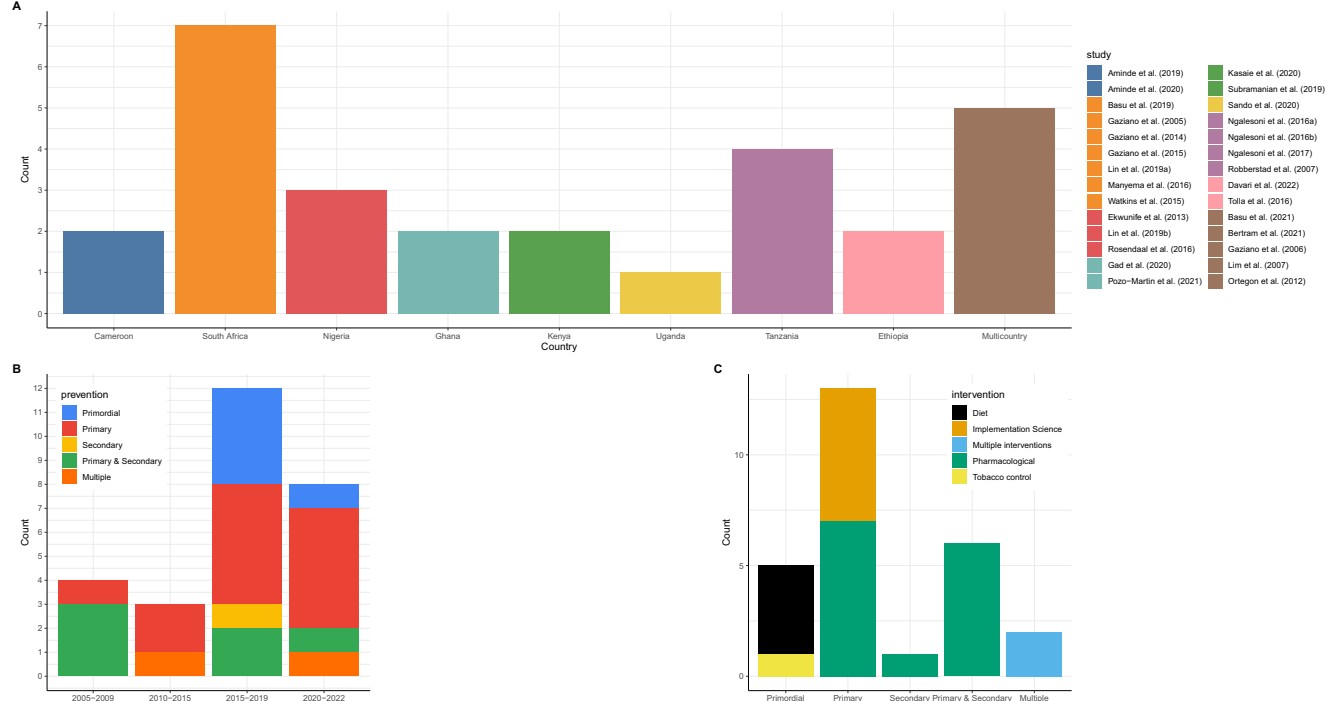

**Fig. 2 | Characteristics of the included studies. A** Distribution of studies by country. Each country has a unique colour, which corresponds with the colour of the studies. Brown colour represents multi-country studies. **B** Distribution of studies by type of prevention and year of publication. Blue colour represents primordial prevention; red for primary prevention; yellow for secondary prevention and green are studies that modelled interventions targeting both primary and secondary prevention. **C** Distribution of studies by type of intervention and level of prevention. The colour codes represent the different types of interventions-black represents diet interventions, yellow for implementation science interventions, blue for studies modelling multiple interventions, green for pharmacological interventions and yellow for interventions targeting tobacco control.

evaluated interventions targeting primordial CVD prevention while eight studies focused on multiple interventions targeting both primary and secondary prevention[24–26,31,35,41,43,47]. One study[36] focused on secondary CVD prevention only (Fig. 2C).

Pharmacological interventions (mainly antihypertensives and statins) were the most evaluated either as single[24–31,35,36,38,39,41,43,46,47] or combined interventions[34,42,44,45]. Six studies[32–34,42,44,45] evaluated implementation science interventions for hypertension screening and treatment. Diet interventions were evaluated in four studies[22,23,37,48] while only one study in Tanzania[40] evaluated interventions targeting tobacco control. Figure 3A and B present the distribution of the evaluated interventions by country.

**Characteristics of the decision analytic models**

Figure 4, 5, and Supplementary Data 1 present the characteristics of the DAMs.

**Types of evaluations and models.** All but three studies[22,39,48] were full economic evaluations involving the comparison of costs and health outcomes of which the majority (23/27) were cost-utility analyses[23–34,36–38,40–47] (Fig. 4A). Thirteen studies were Markov models[27–32,38–44] whereas seven were microsimulation models[24,25,33–36,46]. Markov modelling approach was used by studies evaluating the cost-effectiveness of providing antihypertensive treatment[27–31,38,39,43], multi-component community-based hypertension interventions[42,44], community health worker interventions[32] and tobacco policies[40]. Microsimulation models were used to evaluate the impact of pharmacological interventions[24,25,35,36,46], and multicomponent interventions involving both screening and treatment[33,34]. Three studies used multistate life tables to evaluate the impact of sugar taxation[37] and salt reduction policies[22,23]. The WHO-CHOICE methods were used in three studies to model the impact of multiple interventions[26,41,47] while one study did not specify the model type but reported using an epidemiologic-cost model[45].

(Fig. 4B). In South Africa, four different model types were used while most countries had only one model type (Fig. 5).

**Study perspectives.** Healthcare system perspective of analysis was the most used[23,24,26,29,34,36,37,40,43,46] followed by provider[28,35,44,45,47] and societal perspectives[27,31,38,39,42]. Six studies did not explicitly state the perspective of evaluation[25,30,32,33,41,48], while the perspective was not relevant in one study that focused on health outcomes only[22] (Fig. 4C).

**Time horizon, cycle length and discounting.** The starting age of patients included in 20 models ranged from 15–45 years[24,25,27–33,35–46]. Three studies[22,23,48] modelled whole populations while the starting age of patients was not clear in two studies[26,34,47]. Lifetime horizon was adopted by 17 studies[22–24,26,27,29,31,36–41,43,44,46,47] while eight studies adopted 10–30 year horizons[23,25,28,30,34,35,42,45]. In one study[48], the analyses were performed over one year whereas the horizon was not stated nor clear in two studies[32,33] (Fig. 4D). Annual cycle lengths were the most adopted in 19 studies[22–25,27–30,32–34,36,38–40,42–44,46] while the remaining eight studies did not specify their cycle length[26,31,35,37,41,45,47,48]. None of the studies mentioned performing half-cycle correction. Three percent discount rate was used in all the 22 studies[22–31,33,34,36,38,40–47] where discounting was performed.

**CVD outcomes modelled.** Figure 4F presents the CVD outcomes included in the DAMs. The sum of complications from the graph exceeds the number of studies because all but two studies[26,37] modelled multiple CVD outcomes as health states. Fifteen studies modelled two CVD states[24,27,28,31,35,36,38–45,47], six studies modelled four CVD states[22,23,30,34,46,48], four studies modelled three states[25,29,32,33], while one study modelled only one CVD state[37]. Atherosclerotic CVDs were the commonest health states modelled in all DAMs that specified outcomes, while only six studies[22–25,29,48] included hypertension complications as health states.

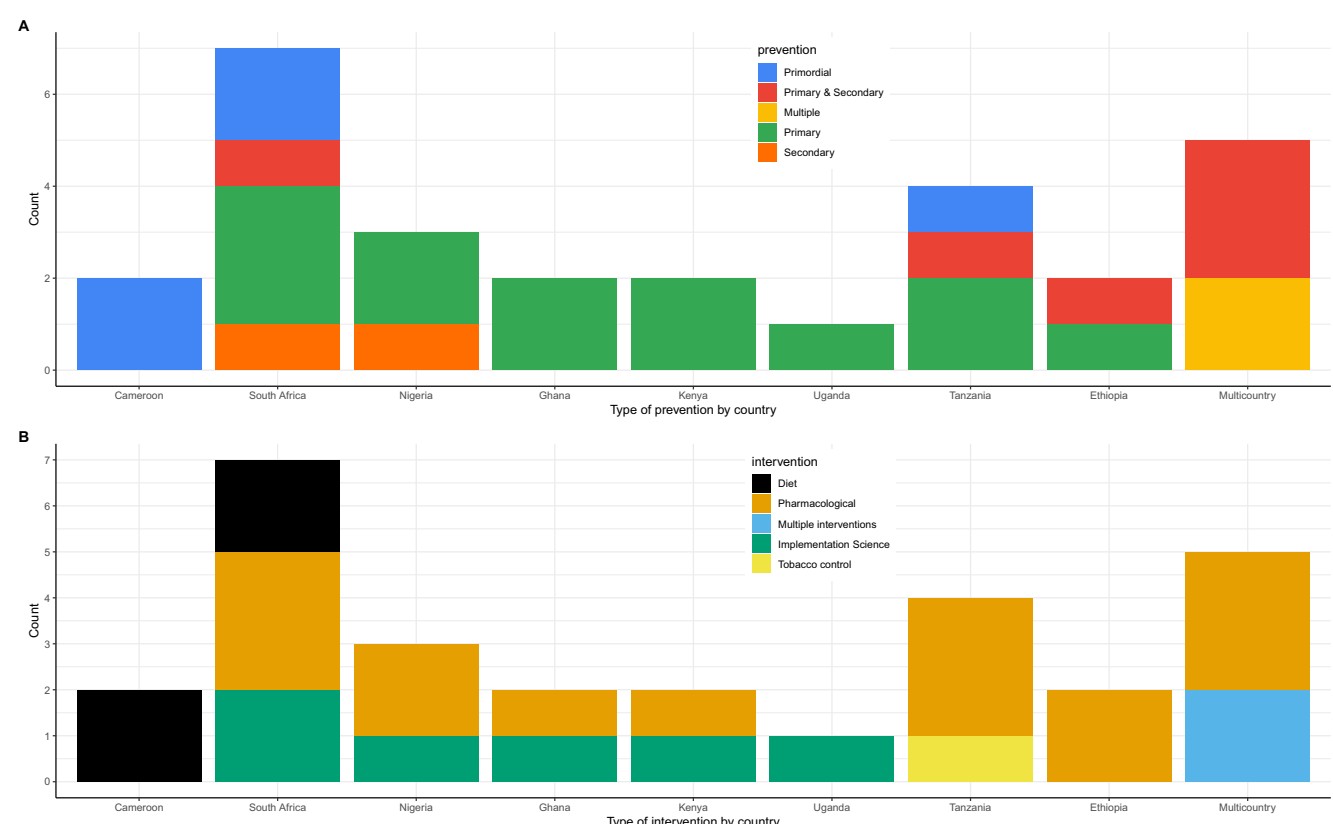

**Fig. 3 | Types of interventions modelled. A** A graph characterizing the level of prevention by country. Each colour uniquely represents a level of prevention. **B** A graph presenting the type of intervention by country. Each colour represents an intervention.

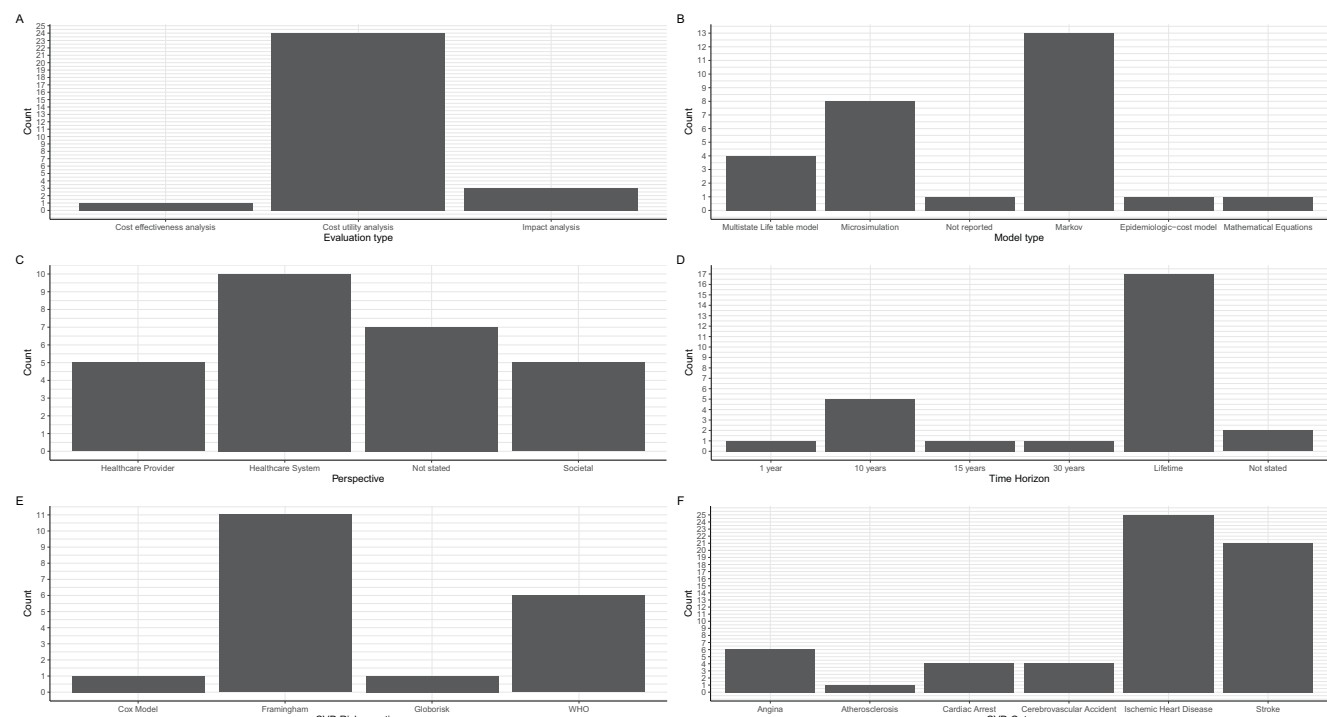

**Fig. 4 | Characteristics of the decision analytic models. A** A graph presenting the type of evaluation performed. **B** A graph showing the type of model used. **C** A graph presenting the study perspective adopted. **D** A graph presenting the time horizon adopted. **E** A graph showing the cardiovascular disease (CVD) risk equation used. WHO stands for World Health Organization. **F** A graph presenting the CVD outcomes modelled.

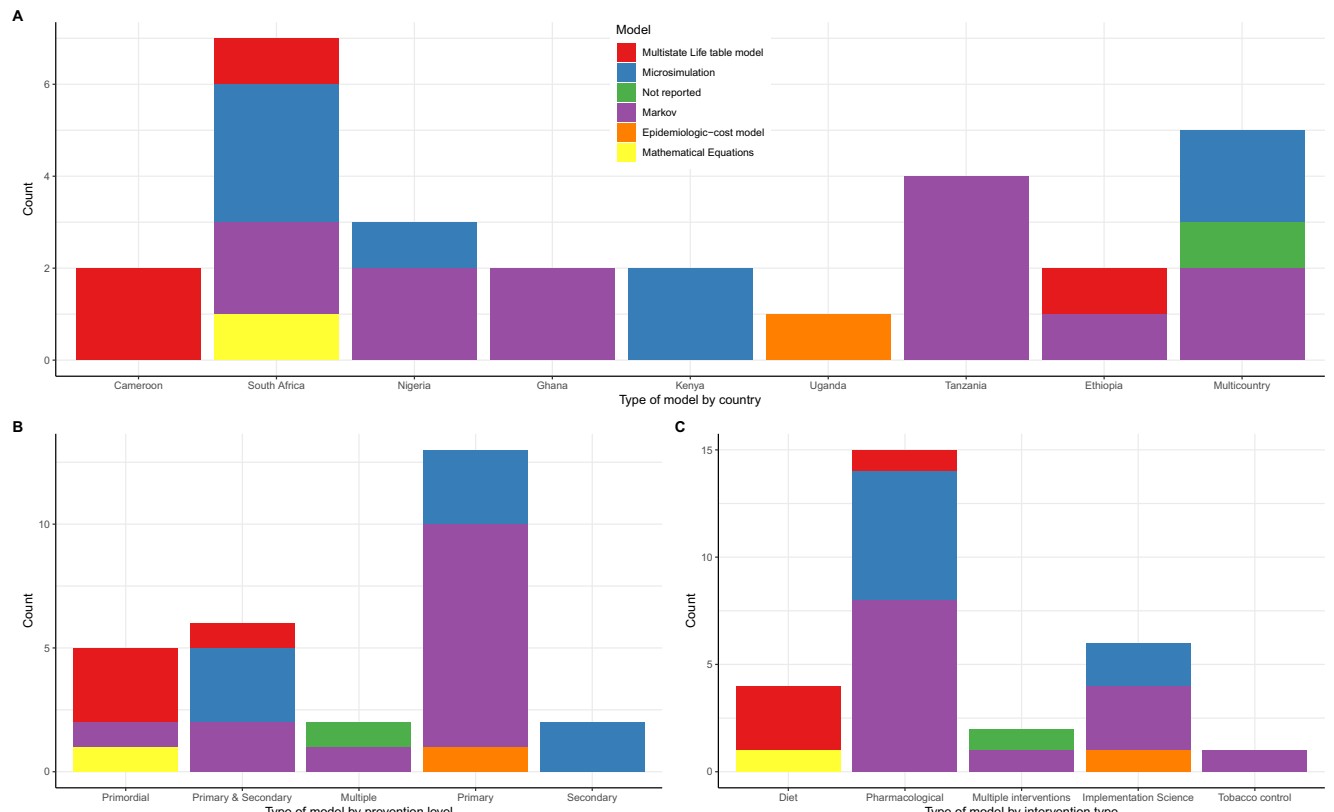

**Fig. 5 | Model type by country, prevention and type of intervention. A** A graph presenting the distribution of model types by country. Each colour is unique to a model type. **B** A graph showing the model type by the level of prevention modelled.

Each colour is unique to a model type. **C** A graph presenting the model type by intervention modelled. Each colour is unique to a model type.

Ischaemic/coronary heart disease and/or stroke were the most common CVD outcomes in all studies except one study[26], which reported CVD as an outcome but did not specify type of CVD (Fig. 4F). Twenty three studies[22–25,27–29,31,32,34,36–48] modelled stroke as an outcome while three studies[30,31,35] included cerebrovascular accidents (CVA). Angina and cardiac arrest were modelled in seven[27,30–34,46] and five[27,30,33,34,46] studies, respectively. Two studies modelled ischaemic and haemorrhagic stroke as separate outcomes[22,23] and also separated hypertensive from ischaemic heart diseases.

**CVD risk equations.** Framingham risk equations were the most used to compute 10-year CVD risk in 11 studies[23,27,28,30–34,38,44,46] (Fig. 4F). Four studies used the World Health Organization (WHO) absolute risk-based approach for computing the 10-year CVD risk[24,25,39,47]. Pozo-Martin et al.[42] used the Framingham risk equation for the base case but performed sensitivity using the WHO CVD risk charts for Western sub-Saharan Africa. Basu et al.[24] used both WHO/International Society of Hypertension (ISH) risk equations and Harvard/National Health and Nutrition Examination Survey (NHANES) to compute CVD risk for patients treated using different guidelines[24]. Gaziano et al.[33] fitted two cox proportional hazards models using the US NHANES 1 dataset to predict the risk for IHD and CVA. In their cost-epidemiologic model, Sando et al.[45] used the Globorisk equations to compute 10-year CVD risk among HIV patients in Uganda.

**Health outcomes and Equity considerations**
Supplementary Data 1 presents the outcome measures included in the models. Majority (18/27) of the studies used disability adjusted life years (DALYs) as the generic measure of the health outcomes[24–27,29,32,34,36–38,40–47]. Four studies used the quality adjusted life years (QALYs)[28,30,31,33] while two studies used the health adjusted life years outcome measures[22,23]. Lim et al[35]. reported deaths averted only. Five studies reported either CVD events or

deaths averted alongside a generic measure of health outcome[23,25,34,36,45]. Robberstad et al.[43] used the life years gained as a surrogate outcome. Seven studies performed different types of equity analyses[22,24,34,37,39,45,48]. Subgroup analysis was used in five studies[22,24,34,37,45], while one study each used extended cost effectiveness analysis (ECEA)[48] and distributional cost effectiveness analysis (DCEA)[39] methodologies. Gender inequalities were the most assessed in four studies that explored the difference in health outcomes between males and females[22,24,37,45]. Three studies[24,39,48] assessed the impact of interventions across different socioeconomic groups. Ngale-soni et al.[39] used life expectancy, Gini coefficient, and achievement index as measures of equity impact of primary CVD prevention. Similarly, Watkins et al.[48] used deaths averted, catastrophic health expenditure averted, and poverty cases averted to measure the equity impact of salt reduction policies in South Africa. Only one study each focused on ethnic[24] and regional inequalities[34].

**Uncertainty and budget impact analyses**
Eighteen studies[22,23,27–32,36–38,40–44,47] performed both one-way and probabilistic sensitivity analyses (PSA) whereas seven studies[24,26,33,45,46,48] performed only one-way sensitivity analyses. One study performed PSA only[28] while two studies[25,39] did not report performing any sensitivity analyses. Seven studies[28,29,34,38,43,44,47] presented cost-effectiveness acceptability curves (CEAC) or frontiers (CEAFs) showing the relative probability of cost-effectiveness of alternative interventions. Only two studies performed value-of-information (VOI) analysis[28,38]. Similarly, only five studies conducted budget impact analyses for the evaluated interventions[24,29,34,36,37].

**Model adaptation and validation**
Five studies adapted previously developed models in international settings to suit their decision problems[27,29,32,33,47]. The CVD policy model, a validated model previously developed for the US population, was adapted to the

Ethiopian[27] and South African[33] settings. In Ghana, one study adapted a 2006 model initially used by the UK NICE to update the hypertension guidelines[29]. Another study[47] adapted the WHO CHOICE model for East Africa to suit the Ethiopian setting. Only three studies reported conducting some form of model validation[27,30,33]. However, the details of the validation were not adequately reported to establish the types of validation performed or the process undertaken. Model calibration was reported in two studies[30,36] while four studies provided details of stakeholder elicitation processes to obtain expert opinion[23,29,42,45].

### Quality assessment based on Philips checklist

Supplementary Data 1 presents the quality appraisal of the included models against the different dimensions of the Philips et al.[17] checklist. The mean quality score of the papers based on the Philips checklist was 68.9% and ranged from 46.4% to 85.1% (median = 72.3%). Fifteen studies scored above 70%, while only two studies scored below 50%. Based on the models' dimensions of quality: the structure dimension scored the highest (84.9%), data dimension averaged 58.0% while the consistency dimension scored the least at 45.8%.

In all the studies, the decision problems were clearly defined and were consistent with the objectives of the evaluations and models specified. However, only 15 studies specified the primary decision maker[22–24,29,30,32,37–40,42,44,46–48]. Fourteen studies did not include all the feasible options in the evaluations[27–29,33,34,36,40,42–48]. The disease states included in almost all the studies reflected the underlying pathophysiology of the disease. Six studies did not define or justify the cycle length[26,37,41,42,45,47].

The data used to construct most models (22/27) were aligned with the objectives of the evaluations. Regarding cost data, 17 studies reported using local sources either from administrative sources or from primary data collection[23,24,27–29,34,37–40,42–48]. However, none of the studies assessed the quality of the data used. Almost half of the studies (12/27) did not justify the choices made between different data sources[26,28,30–33,39–43,46]. The majority of the studies did not report the processes used to elicit expert opinion (21/27). None of the studies performed all the four principal types of uncertainty analyses (methodological, structural, heterogeneity, and parameter). Parameter uncertainty was the most assessed through sensitivity analyses while structural uncertainty was the least addressed.

Nine studies reported performing tests of the mathematical logic of the model before use. However, only two studies[30,36] reported performing model calibration against independent data, but the details were very scanty. The majority of the studies (21/27) compared their results with those of previous models.

## Discussion

We included a total of 27 studies in this systematic review from eight SSA countries. The majority of the studies were published after 2015 and focused on pharmacological interventions, with the fewest number focusing on lifestyle interventions for CVD prevention. There was heterogeneity in the modelling methods used with Markov models being the most used to evaluate the impact of CVD prevention. The most captured CVD outcomes were ischaemic heart disease and stroke. Framingham CVD risk equations were the most used to predict the 10-year CVD risk for patients included in the model. Lifetime horizon was the most adopted, but some studies used shorter time horizons. Gender and socioeconomic dimensions were the most examined by the equity-focused studies. The majority of the studies had a high mean quality score, but consistency and data dimensions scored the least. Data limitations, especially for key parameters like treatment effect and CVD risk, were recurrent themes across most studies.

Consistent with previous reviews[13–15], this review found that most studies focused on primary CVD prevention, with the majority evaluating pharmacological interventions especially antihypertensives. It is not surprising that antihypertensives were the most evaluated intervention given the high burden of hypertension in SSA, which affects almost half of the population aged above 25 years and has a significant impact on household incomes[49,50]. Despite the high prevalence, only about a quarter (27%) of the

hypertensive individuals in SSA are aware about their status, 18% are on treatment, and a paltry 7% attaining blood pressure control[51]. In this review, only six studies[32–34,42,44,45] evaluated different primary healthcare interventions for hypertension screening and management. Stronger primary healthcare (PHC) systems have been identified as the most feasible way towards the attainment of UHC and other health-related SDGs[52]. It is important to evaluate alternative PHC approaches that can be implemented to increase the coverage of CVD prevention interventions, especially among the unreached populations in SSA. This includes identifying different population groups that would be impacted by the interventions by examining the health and financial risk impacts.

Interventions targeting primordial prevention, specifically behavioural risk factors, in SSA, were the least evaluated. For instance, only one study evaluated tobacco interventions in Tanzania[40] while salt[22,23,48] and sugar[37] interventions were evaluated only in two countries (South Africa and Cameroon). Lifestyle interventions fall within the 'WHO best buys' and their implementation can significantly reduce the onset of CVDs in SSA. Evidence shows that about 81% of adults in SSA consume more than the recommended 2 g sodium per day[53] and that SSA has experienced the highest rise in sugar-sweetened beverage (SSB) consumption compared to other regions[54]. For SSA to significantly reduce the CVD burden, it is imperative that there is sustained focus towards primordial prevention, which requires health economic evidence to inform decision-making.

We observed an increasing number of model-based studies since 2010, with almost three-quarters of the studies being published after 2015. Similarly, we observed an increasing number of prevention interventions being evaluated, especially after 2015. This can be attributed to increased global commitments to meeting CVD prevention and control targets by 2025[55], the UN sustainable development goals[56], and enhanced collaboration within and without the region[57]. Governments and other stakeholders in SSA increasingly recognize the need for using economic evidence in the design of health benefit packages, especially with the quest towards attaining UHC[58]. However, given the diversity within the African continent and differences in settings, additional modelling studies are required for context-specific evidence that can inform priority setting in individual countries.

Conceptual modelling and model selection processes were poorly documented despite modelling approaches being aligned to the decision problem of interest. Markov models, microsimulations and multi-state cohort life table models were the most used methods. Previous reviews found that Markov models were the commonest modelling methods in LMICs[13,16]. The multistate cohort life table modelling approach was adopted mainly by studies modelling whole populations to examine the impact of salt and sugar policies on multiple diseases in Cameroon[22,23] and South Africa[37]. Compared to cohort-based approaches that model aggregate populations, individual patient level models follow individual trajectories as they experience events of interest and average their costs and outcomes to derive population averages. Individual patient level models permit the modelling of patient heterogeneity and suit complex interventions[59] but are also data hungry and computationally intensive. The trade-off between different modelling methods depends on the nature of the decision problem, data availability and resources. It is important for modellers to conduct and properly document the conceptual modelling process to inform the model selection process.

The review found that only seven studies incorporated equity dimensions in their analyses[22,24,34,37,39,45,48], of which five performed subgroup analyses while only two[40,48] used generic equity metrics. Gender, age, and socioeconomic dimensions were the most explored, while only one study each examined the differential impact of interventions on ethnicities[24] and regions[34]. A review in LMICs reported an increasing focus on equity analysis in recent economic evaluations[60]. Only two studies in our review used ECEA[48] or DCEA[39] methodologies to undertake their equity analyses. While most equity-focused studies perform subgroup analyses, newer methods like extended (ECEA) and distributional (DCEA) cost-effectiveness analyses are being adopted to undertake equity focused economic evaluation[60]. However, these methods have not been extensively applied in existing

DAMs for CVD prevention in SSA. Incorporating equity dimensions in economic evaluations of CVD prevention is particularly relevant to the SSA context considering the need to scale up intervention coverage targeted at various population groups while at the same time ensuring that the financial barriers to accessing healthcare are eliminated.

Whereas most studies used country-specific data sources to inform baseline population and cost parameters, critical data gaps were observed relating to intervention effectiveness and CVD risk equations. There was a lack of local data for generating 10-year CVD risk equations relevant to the SSA context. Where 10-year CVD risks were estimated, the Framingham risk equations and WHO/ISH risk prediction charts were the most common approaches. In a few cases, the Globorisk algorithm and cox proportional hazards models were fitted using data from other settings. All the CVD risk prediction models differ in terms of their sensitivity and hence may underestimate or overestimate the risk of CVD in a particular population[61,62]. This review highlights the need for longitudinal studies in SSA, especially cohort studies, that involve long-term follow-up of patients with different risk profiles to better understand the natural history and probability of developing CVDs. Another critical data gap relates to the utility values used to compute QALYs gained from alternative interventions. For instance, all the four studies[28,30,31,33] that used QALYs as health outcome measures derived their utility values from developed country settings. This finding calls for individual countries in SSA to invest in health valuation studies using multi-attribute utility instruments like the EQ5D so as to generate local value sets that can be used to compute QALYs for future modelling studies.

Despite a high overall quality score, we observed heterogeneity in the methods applied in modelling CVD prevention interventions in SSA. While there exist different health economic evaluation guidelines[17,63–69], we used the Philips checklist[17] due to its suitability in assessing the quality of DAMs. The model structure dimension scored the highest while the consistency dimension scored the least. Most studies did not report evaluating the quality of data included in the models, consistent with the findings from previous review[16]. Uncertainty analyses were also not adequately performed in some models, with structural uncertainty being the least addressed. While VOI analysis can be useful to quantify uncertainty and better inform decision-makers, none of the studies performed VOI analysis. In addition, model validation and calibration were rarely done and where done, scantily reported. Stakeholder engagement and elicitation processes were also not adequately reported in most models. It is imperative that modellers consider effective stakeholder engagement during the modelling process to inform the assumptions, and enhance transparency and use of the evidence[70]. Model validation guidelines[70] should be adhered to in order to promote model accuracy and stakeholder confidence. Given the resource constraints in SSA, it is important not only to rely on cost-effectiveness but budget impact of interventions. However, the majority of the studies did not perform budget impact analysis, which does not provide a comprehensive picture about the consequences of adopting new interventions. All the included studies used a 3% discounting rate, but some studies did not perform any sensitivity analysis to assess the effect of varying the discounting rate on the results. Haacker and colleagues[71,72] recommend the use of a discounting rate of at least 5% for low and lower-middle income countries and 4% for upper-middle income countries. At the very least, modellers should conduct sensitivity analyses around the discounting rate to assess the effect of different rates on the result. These findings highlight the need for modellers in SSA to adhere to best practices while building their DAMs. As much as possible, DAMs should be relevant to the context and should use local data to ensure that the analyses are useful to the setting. Modellers should also ensure that they assess the different types of uncertainty to test the robustness of their results under different scenarios.

This review has some limitations. We only included articles published in the English language and also did not include grey literature which could exist outside the academic databases searched. Moreover, the heterogeneity in the interventions and modelling types made model comparisons unfeasible. Nevertheless, the review provides a comprehensive picture on the application of DAMs for evaluating interventions targeted at CVD prevention in SSA.

## Conclusion

This systematic review provides an overview of the existing literature on model-based economic evaluations of interventions targeting CVD prevention in SSA. The review finds a paucity of studies modelling the impact of primordial prevention interventions and those targeting the scale up of screening and treatment of CVD risk factors to prevent CVD onset, especially among the undiagnosed but high-risk individuals in SSA. Appropriate modelling methods should be used for complex interventions, especially those with heterogeneity and interactions. Moreover, there is a need to explore equity dimensions in economic evaluations of CVD prevention in order to expand intervention coverage and reach the significant proportion of the SSA population without access. The review also highlights the need for longitudinal studies in SSA to facilitate more appropriate CVD risk prediction and for local and context specific health outcome valuation studies. Modellers should adhere to modelling best practices and improve their transparency in model building, validation, documentation.

## Data availability

This systematic review is based on data extracted from studies published in publicly available literature. All data generated or analysed during this study are included in this published article and its figures and supplementary files. The source data is located in Supplementary Data 1.

## Code availability

The R code for reproducing the figures is stored on GitHub[73].

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

## Acknowledgements

The authors thank Ms. Louise Falzon, information specialist at the Sheffield Centre for Health and Related Research (SCHARR), Division of Population Health, University of Sheffield library for the critical inputs during the development and review of the search strategy. This work was funded by the Wellcome Trust as part of a doctoral training grant [218462/Z/19/Z] awarded to JOO to pursue PhD in Public Health Economics and Decision Science at the University of Sheffield. The funders had no role in the design of the study, data collection, analysis, and interpretation, or in writing the manuscript.

## Author contributions

Concept and design: JO, PB, PJD. Drafting of review protocol: JO with inputs from PB and PJD. Literature Searching: JO. Abstract and full text screening: JO, EW, PK and CA. Extraction of data: JO, EW, PK, and CA. Analysis and interpretation of data: JO, PB and PJD. Drafting of the manuscript: JO, with inputs from PB and PJD. Critical revision of the paper for important intellectual content: JO, EW, PK, CA, PO, PB, and PJD. Obtaining funding: PB, PJD. Administrative, technical, or logistic support: JO, PB, PJD. Supervision: PB, PJD.

## Competing interests

The authors declare no competing interests.
