## [Transparent Peer Review file · Communications Medicine]

Application of Decision Analytic Modelling to Cardiovascular Disease Prevention in Sub-Saharan Africa: A Systematic Review

Corresponding Author: Mr James Oguta

Version 0:

Reviewer comments:

Reviewer #1

(Remarks to the Author)

The authors of this paper highlight the high burden of cardiovascular diseases (CVDs) in low- and middle-income countries (LMICs) as a major healthcare concern. There is a need to both adopt and scale up effective primordial, primary and secondary prevention interventions. Decision makers often rely on decision analytic modelling (DAM) to evaluate the health, economic and equity impact of different interventions for CVD prevention. The authors evaluate the use on DAMs in SSA by appraising the characteristics and quality of existing DAMs, the types of prevention interventions modelled, how CVD progression was modelled, as well as how equity considerations were incorporated. The results are relevant to SSA and potentially other LMICs.

Thank you for the opportunity to review this well written review. This is an important piece of work that is likely to be crucial for improving the use of DAMs in SSA and LMICs as a whole. The Introduction and the features of DAMs are very clear even for non-specialists. The methodology is appropriate and well-described, and the results are comprehensively presented. The discussion of the selected studies as a whole is also clear and correct. Nevertheless, I have a few questions and observations:

1. The authors mention the use of a 3% discount rate for all the studies that applied discounting. Lately there has been some discourse on the appropriateness of using 3% as a discount rate in LMICs. There might be some benefit in discussing the appropriateness of using this discount rate in future studies.
2. Page 14- line 359: The abbreviation SSB has not been previously defined.
3. Page 14- Line 378: Remove each?
4. Page 14- Line 378: Sentence is important but seems like an afterthought. Consider rephrasing this sentence to improve flow.
5. Page 15- Line 382: Statement not very clear. Are there any studies from the selected studies that applied ECEA/DCEA's? If not consider removing extensively as it suggests partial use of the methodologies.
6. Consistency when abbreviating. There are instances where the full words even after introducing the abbreviations. Example; Page 16, Line 416: DAMs.
7. You highlight the use (or lack thereof) of budget impact analysis and value-of-information (VOI) analysis in the results. It is probably worthwhile discussing the importance of using these methodologies alongside CEAs/DAMs particularly in SSA and LMICs.
8. The Include Box is not showing clearly in Figure 1
9. Based on Figure 2, it would be beneficial to highlight the increase in studies involving DAMs over the years, as well as the growth in the variety of prevention types being studied over time.
10. Based on Figure 4, consider explicitly stating the studies that modelled multiple CVD outcomes(explaining why the sum of counts is more than 27) and if any only modelled only one outcome. This information is not available in the tables so it might be useful to mention it in your results.

Reviewer #2

(Remarks to the Author)

This paper by Oguta and colleagues addresses the topic of decision-analytical models for CVD in sub-Saharan Africa. Because CVD is a growing problem in the region but resources are limited, these models are an important part of priority-setting and policy development. This is an important and timely review.

I have a few comments for the authors' consideration:

The decision to exclude certain types of CVD needs to be better justified. I suggest listing the outcomes of interest and the rationale. Most of the studies covered here address atherosclerotic CVD (ischemic heart disease + ischemic stroke) and/or hemorrhagic stroke. If that is what the authors mean by CVD then just say that. But what about heart failure, degenerative valve disease, congenital heart disease, etc? These are all "other" CVDs that are not rheumatic heart disease but appear to have been excluded. Table 1 PICO is confusing because these other CVDs are neither explicitly included or excluded. Also "stroke" and "cerebrovascular accident" are the same thing.

I would have expected an analysis of the structure of the models themselves with some reflections on authors' choice of health states and outcomes vis a vis their research questions. For instance, a primordial prevention paper doesn't need to get into the nuances different CVD complications (heart failure, repeat MI, etc), whereas a secondary prevention paper probably does. What are the standards for a 'good' CVD model in the African context?

What does the term "health systems strengthening" mean in the context of this paper? The four studies that are labeled as HSS were all looking at task-sharing or integrated models of care delivery. These are more akin to implementation research than to HSS itself, which is usually focused on improving the level and allocation of health system inputs rather than new ways to deliver evidence-based interventions.

Did the studies that use QALYs conduct their own surveys (using MAUIs) and use local value sets to get QALY weights? If not, that would be important to point out because this is a pervasive problem in the global health economics literature.

The reality is that most researchers design their studies after previous studies, so in many cases they may simply be replicating the mistakes of previous studies and may not be aware of all the elements of the Philips checklist. If the objective of this paper is in part to improve the quality and validity of future CVD decision analyses in the region, then it would be advisable to have a table or figure in the Discussion section that gives some recommendations on how to improve CVD modeling studies going forward.

Finally, given that this study uncovered a range of diverse papers with disparate objectives, and there was no primary quantity of interest or meta-analysis done, I would suggest reframing this as a scoping review rather than a systematic review. The existing SR elements can be readily adapted to the PRISMA-P checklist.

Version 1:

Reviewer comments:

Reviewer #1

(Remarks to the Author)

Thank you for the responses to the concerns raised.

Reviewer #2

(Remarks to the Author)

The authors have done a good job of responding to my original review. I am satisfied with the revision and have no further comments.

RESPONSE TO REVIEWERS

Referee expertise:

Referee #1: epidemiology and biostats

Referee #2: Global health; Health economics; epidemiology

Reviewers' comments:

Reviewer 1:

Reviewer #1 (Remarks to the Author):

The authors of this paper highlight the high burden of cardiovascular diseases (CVDs) in low- and middle-income countries (LMICs) as a major healthcare concern. There is a need to both adopt and scale up effective primordial, primary and secondary prevention interventions. Decision makers often rely on decision analytic modelling (DAM) to evaluate the health, economic and equity impact of different interventions for CVD prevention. The authors evaluate the use of DAMs in SSA by appraising the characteristics and quality of existing DAMs, the types of prevention interventions modelled, how CVD progression was modelled, as well as how equity considerations were incorporated.

The results are relevant to SSA and potentially other LMICs.

Thank you for the opportunity to review this well written review. This is an important piece of work that is likely to be crucial for improving the use of DAMs in SSA and LMICs as a whole. The Introduction and the features of DAMs are very clear even for non-specialists. The methodology is appropriate and well-described, and the results are comprehensively presented. The discussion of the selected studies as a whole is also clear and correct.

We thank the reviewer for taking time to review the paper, finding our work a significant contribution to knowledge, and giving very valuable feedback that helps to improve the quality of the paper. We have considered each of the comments and responded as follows:

Nevertheless, I have a few questions and observations:

1. The authors mention the use of a 3% discount rate for all the studies that applied discounting. Lately there has been some discourse on the appropriateness of using 3% as a discount rate in LMICs. There might be some benefit in discussing the appropriateness of using this discount rate in future studies.

Thank you for this comment. We acknowledge the ongoing discourse regarding the use of a 3% discount rate for LMICs ^{1,2}, especially the argument by Haacker and colleagues ¹ recommending the use of a discounting rate of at least 5% for low and lower-middle income countries and 4% for upper-middle income countries. We have now updated the discussion section to include the discounting rate (Lines 419-426).

2. Page 14- line 359: The abbreviation SSB has not been previously defined.

Thanks for this observation. We have now defined SSB as “sugar sweetened beverages”

3. Page 14- Line 378: Remove each?

Thank you. We have revised the sentence to improve its flow.

4. Page 14- Line 378: Sentence is important but seems like an afterthought. Consider rephrasing this sentence to improve flow.

Thanks a lot for the comment. We have revised the sentence to improve its flow.

5. Page 15- Line 382: Statement not very clear. Are there any studies from the selected studies that applied ECEA/DCEA's? If not, consider removing extensively as it suggests partial use of the methodologies.

Thank you for the comment. As mentioned in Lines 271-272, there were two studies that used ECEA and DCEA. We have added a preceding sentence before it to remind the reader about the finding. (See Line 378-379)

6. Consistency when abbreviating. There are instances where the full words even after introducing the abbreviations. Example; Page 16, Line 416: DAMs.

Thanks for the comment. We have reviewed the entire manuscript and corrected inconsistencies in abbreviations as advised.

7. You highlight the use (or lack thereof) of budget impact analysis and value-of-information (VOI) analysis in the results. It is probably worthwhile discussing the importance of using these methodologies alongside CEAs/DAMs particularly in SSA and LMICs.

Thank you for this comment. We have added the discussion on the relevance of budget impact and VOI analysis to LMICs (See Line 410)

8. The Include Box is not showing clearly in Figure 1

Thanks for the observation. We have edited the box to make “Included” more visible.

9. Based on Figure 2, it would be beneficial to highlight the increase in studies involving DAMs over the years, as well as the growth in the variety of prevention types being studied over time.

Thank you for the comment. We have now updated the discussion to highlight the increasing number of studies over time as well as the growth in the variety of prevention types being studied over time. (See Lines 350-359)

10. Based on Figure 4, consider explicitly stating the studies that modelled multiple CVD outcomes(explaining why the sum of counts is more than 27) and if any only modelled only one outcome. This information is not available in the tables so it might be useful to mention it in your results.

Thank you for the observation. We have now updated the results section to explain the figure further and explicitly classified the studies by the number of CVD health states modelled. (See Lines 223-229)

Reviewer 2:

Reviewer #2 (Remarks to the Author):

This paper by Oguta and colleagues addresses the topic of decision-analytical models for CVD in sub-Saharan Africa. Because CVD is a growing problem in the region but resources are limited, these models are an important part of priority-setting and policy development. This is an important and timely review.

We thank the reviewer for dedicating time to critically appraise our manuscript, for highlighting the importance and timeliness of the review and for providing very useful feedback that have helped improve the quality of the paper. We have considered all the comments and suggestions, revised the manuscript appropriately and responded as follows:

I have a few comments for the authors' consideration:

The decision to exclude certain types of CVD needs to be better justified. I suggest listing the outcomes of interest and the rationale. Most of the studies covered here address atherosclerotic CVD (ischemic heart disease + ischemic stroke) and/or hemorrhagic stroke. If that is what the authors mean by CVD then just say that. But what about heart failure, degenerative valve disease, congenital heart disease, etc? These are all "other" CVDs that are not rheumatic heart disease but appear to have been excluded. Table 1 PICO is confusing because these other CVDs are neither explicitly included or excluded. Also "stroke" and "cerebrovascular accident" are the same thing.

Thank you for the comment. We performed a thorough literature search, and rigorous screening and study selection, and to the best of our knowledge, we did not miss any study that modelled adult CVDs in SSA. Heart failure and valvular diseases were included in our search strategy. In addition, our search string included the terms "cardiovascular disease" or "heart disease" ... or "CVD", which limits the possibility of omitting a CVD focused paper. As explained in the manuscript (Lines 113- 116), we only focused on CVDs that affect adult populations, justifying the exclusion of congenital heart diseases and RHD, which predominantly affect children. We have updated table 1 to explicitly state that non-atherosclerotic CVDs were included and congenital heart diseases excluded, and reported the studies that included non-atherosclerotic CVDs in the results section (Lines 223-229).

The search strings below (From MEDLINE database search and in the Appendix) show how we constructed the CVD search term, to include all the important CVDs:

8 ("cardiovascular disease" or "heart disease" or stroke or "myocardial infarction" or
"myocardial ischaemia" or "transient ischemic attack" or "ischemic attack" or "cerebrovascular
disease" or "cerebrovascular accident" or CVA or IHD or CVD or CHD or "cardiovascular event"
or angina or "angina pectoris" or "heart attack" or "ischemic heart disease*" or "coronary heart
disease" or "coronary disease" or "heart failure" or "acute coronary syndrome" or "peripheral
vascular disease" or "Peripheral Vascular diseases" or "atrial fibrillation").tw. 1063014
9 exp Heart Diseases/ 1276479
10 exp Cardiovascular Diseases/ 2735336
11 8 or 9 or 10 3030748

I would have expected an analysis of the structure of the models themselves with some reflections on authors' choice of health states and outcomes vis a vis their research questions. For instance, a primordial prevention paper doesn't need to get into the nuances of different CVD complications (heart failure, repeat MI, etc), whereas a secondary prevention paper probably does. What are the standards for a 'good' CVD model in the African context?

Thank you for these comments. We agree that the decision problem can and should influence the type and structure of the model. Because our search was defined by methodology, and did not seek to identify modelling studies with a homogeneous set aims, and because modelling study aims influence model structure, this makes it very challenging to legitimately compare model structures across these disparate studies. However, it is more straightforward to classify model type, and in response to your suggestion we have included a new analysis in Figure 5 that shows what types of model have been used by intervention type, country and prevention type.

In terms of your question about what makes a 'good' CVD model in the African context, we would say it is the same things that make a good CVD model in any other context. It was not our goal to develop yet more guidance on good practice in developing and reporting modelling studies. However, in response to your comment we have emphasised the need to adhere to good modelling practices while undertaking DAMs in SSA, and the need for more data from SSA to inform any differences in natural history and data to generate local EQ5D value sets relevant to the setting (see Lines 397-402 and Lines 416-426)

Figure 5: Model type by intervention/prevention type and country

What does the term "health systems strengthening" mean in the context of this paper? The four studies that are labelled as HSS were all looking at task-sharing or integrated models of care delivery. These are more akin to implementation research than to HSS itself, which is usually focused on improving the level and allocation of health system inputs rather than new ways to deliver evidence-based interventions.

Many thanks for your feedback. We agree that the term HSS can be ambiguous as acknowledged by previous studies^{3,4}. As advised, have now revised the paper and classified the interventions as "implementation science" interventions as advised.

Did the studies that use QALYs conduct their own surveys (using MAUIs) and use local value sets to get QALY weights? If not, that would be important to point out because this is a pervasive problem in the global health economics literature.

Many thanks for this feedback. We can confirm that none of the four studies that used QALYs as an outcome measure conducted their own surveys nor used local value sets. We have pointed this out and updated the discussion to highlight the need for conducting local studies for better measurement of health outcomes (See lines 397-402).

The reality is that most researchers design their studies after previous studies, so in many cases they may simply be replicating the mistakes of previous studies and may not be aware of all the elements of the Philips checklist. If the objective of this paper is in part to improve the quality and validity of future CVD decision analyses in the region, then it would be

advisable to have a table or figure in the Discussion section that gives some recommendations on how to improve CVD modelling studies going forward.

Many thanks for this comment. We agree that many researchers do rely on previous studies to inform the design of their studies, which increases the likelihood of replicating mistakes. However, as mentioned above, it was not our goal in this paper to develop additional guidance on model development, economic evaluation, or their reporting. In light of your comment though, we have updated the discussion section to highlight these observations, and the existing guidelines that are available ⁵⁻¹² and recommended the need for modellers to adhere to the guidelines in order to improve the quality of DAMs in SSA (See Lines 404-406, Lines 410-411 and Lines 416-426).

Finally, given that this study uncovered a range of diverse papers with disparate objectives, and there was no primary quantity of interest or meta-analysis done, I would suggest reframing this as a scoping review rather than a systematic review. The existing SR elements can be readily adapted to the PRISMA-P checklist.

Thank you for this comment. In light of your comment, we consulted Prof Andrew Booth (an expert in review methodology and author of “Systematic Approaches to a Successful Literature Review” ¹³) on the most appropriate terminology to describe our study. His feeling was that the disparate objectives of the paper was not relevant, because our aim was to identify studies with a common methodology (DAMs) across the general area of CVD prevention. According to the Cochrane handbook ¹⁴, a systematic review “attempts to collate all the empirical evidence that fits a pre-specified eligibility criteria to answer a specific question.” and should be explicit, systematic, transparent, and reproducible in methodology. Moreover, the research question should be specified a priori, clear in terms of scope and the inclusion criteria should be clearly defined. Finally, the included studies should be analysed objectively in an impartial manner and conclusions drawn to answer the predefined research questions ¹⁴. By contrast scoping reviews are usually conducted to obtain general patterns in broader literature and mainly as a precursor to systematic reviews ¹⁵. A scoping review addresses broader research questions and is usually conducted with the aim of mapping literature relevant to answering the review question ¹⁶. In addition, scoping reviews do not involve assessing the quality of the included studies.

In view of this, we feel our pre-registered review is more appropriately described as a systematic review (adhering to the PRISMA-P checklist ¹⁷) and not a “Scoping Review” (conducted according to PRISMA-ScR guideline ¹⁸).

References

1. Haacker, M., Hallett, T. B. & Atun, R. On discount rates for economic evaluations in global health. *Health Policy Plan.* **35**, 107–114 (2020).
2. Cohen, J. T. It is time to reconsider the 3% discount rate. *Value Health* **27**, 578–584 (2024).
3. Swanson, R. C. *et al.* Toward a consensus on guiding principles for health systems strengthening. *PLoS Med.* **7**, e1000385 (2010).
4. Witter, S. *et al.* Health system strengthening-Reflections on its meaning, assessment, and our state of knowledge. *Int. J. Health Plann. Manage.* **34**, e1980–e1989 (2019).
5. Philips, Z., Bojke, L., Sculpher, M., Claxton, K. & Golder, S. Good practice guidelines for decision-analytic modelling in health technology assessment: a review and consolidation of quality assessment. *Pharmacoeconomics* **24**, 355–371 (2006).
6. Philips, Z. *et al.* Review of guidelines for good practice in decision-analytic modelling in health technology assessment. *Health Technol. Assess.* **8**, (2004).
7. Husereau, D. *et al.* Consolidated Health Economic Evaluation Reporting Standards 2022 (CHEERS 2022) statement: Updated reporting guidance for health economic evaluations. *Pharmacoeconomics* **40**, 601–609 (2022).
8. Wilkinson, T. *et al.* The international decision support initiative (iDSI) reference case for health economic evaluation. Preprint at <https://doi.org/10.7490/F1000RESEARCH.1116869.1> (2019).
9. Sanders, G. D. *et al.* Recommendations for conduct, methodological practices, and reporting of cost-effectiveness analyses: Second Panel on cost-effectiveness in health and Medicine. *JAMA* **316**, 1093–1103 (2016).
10. Drummond, M. F. & Jefferson, T. O. Guidelines for authors and peer reviewers of economic submissions to the BMJ. *BMJ* **313**, 275–283 (1996).
11. Du, K. J. *et al.* Prof. Michael F. drummond: Reporting guidelines for health economic evaluations: BMJ guidelines for authors and peer reviewers of economic submissions. *Ann. Transl. Med.* **10**, 842 (2022).

12. Watts, R. D. & Li, I. W. Use of checklists in reviews of health economic evaluations, 2010 to 2018. *Value Health* **22**, 377–382 (2019).
13. Andrew Booth, Anthea Sutton, Mark Clowes and Marrison Martyn-St James. *Systematic Approaches to a Successful Literature Review*. (SAGE Publications, Thousand Oaks, CA, 2022).
14. Toby J Lasserson, James Thomas, Julian PT Higgins. Chapter 1: Starting a review. in *Cochrane Handbook for Systematic Reviews of Interventions* (ed. Julian P.T. Higgins, James Thomas, Jacqueline Chandler, Miranda Cumpston, Tianjing Li, Matthew J. Page and Vivian A. Welch) 3–4 (John Wiley & Sons, Chichester, England, 2019).
15. Munn, Z. *et al.* Systematic review or scoping review? Guidance for authors when choosing between a systematic or scoping review approach. *BMC Med. Res. Methodol.* **18**, 143 (2018).
16. Peters, M. D. J. *et al.* Guidance for conducting systematic scoping reviews. *Int. J. Evid. Based Healthc.* **13**, 141–146 (2015).
17. Shamseer, L. *et al.* Preferred reporting items for systematic review and meta-analysis protocols (PRISMA-P) 2015: elaboration and explanation. *BMJ* **350**, g7647 (2015).
18. Tricco, A. C. *et al.* PRISMA extension for Scoping Reviews (PRISMA-ScR): Checklist and explanation. *Ann. Intern. Med.* **169**, 467–473 (2018).